# Seroprevalence and risk factors for Q fever and Rift Valley fever in pastoralists and their livestock in Afar, Ethiopia: A One Health approach

**Regina Bina Oakley** [1,2]*, **Gizachew Gemechu**[3], **Ashenafi Gebregiorgis**[3], **Ayinalem Alemu**[3], **Jakob Zinsstag**[2,4], **Daniel Henry Paris**[1,2], **Rea Tschopp**[2,3,4]

**1** Department of Medicine, Swiss Tropical and Public Health Institute, Allschwil, Switzerland, **2** University of Basel, Basel, Switzerland, **3** One Health Division, Armauer Hansen Research Institute, Addis Ababa, Ethiopia, **4** Department of Epidemiology and Public Health, Swiss Tropical and Public Health Institute, Allschwil, Switzerland

* regina.oakley@swisstph.ch

**Data Availability Statement:** The authors confirm that all data underlying the findings are fully available without restriction. All relevant data are

## Abstract

### Background

*Coxiella burnetii*, the causative agent of Q fever, and Rift Valley fever virus are two under-researched zoonotic pathogens in Ethiopia. Potential outbreaks of these diseases, in light of the high dependency of nomadic pastoralists on their livestock, poses a risk to both human and animal health in addition to risking the pastoralists livelihoods. Our study aimed to determine the seroprevalence and associated risk factors for Q fever and Rift Valley fever in pastoral communities in the Afar region of north-eastern Ethiopia.

### Methodology/Principal findings

This cross-sectional study screened pastoralists (n = 323) and their livestock (n = 1377) for IgG antibodies to *Coxiella burnetii* and Rift Valley fever virus. A seroprevalence for Q fever of 25.0% (95%CI 18.6–32.6) was found in pastoralists and 34.3% (95%CI 27.9–41.3) in livestock overall; with 51.9% in goats (95%CI 44.9–58.8), 39.9% in sheep (95%CI 24.6–51.2), 16.3% in camels (95%CI 10.4–24.6) and 8.8% in cattle (95%CI 5.0–15.0). For Rift Valley fever the seroprevalence in pastoralists was 6.1% (95%CI 3.3–11.0) and 3.9% (95%CI 2.6–5.7) in livestock overall; cattle had the highest seroprevalence (8.3%, 95%CI 3.3–19.2), followed by goats (2.7%; 95%CI 1.4–5.1), sheep (2.5%; 95%CI 1.0–5.9) and camels (1.8%; 95%CI 0.4–6.9). Human Q fever seropositivity was found to be associated with goat abortions (OR = 2.11, 95%CI 1.18–3.78, p = 0.011), while Rift Valley fever seropositivity in livestock was found to be associated with cattle abortions (OR = 2.52, 95%CI 1.05–6.08, p = 0.039).

### Conclusions/Significance

This study provides evidence for a notable exposure to both Q fever and Rift Valley fever in pastoralists and livestock in Afar. The outbreak potential of these pathogens warrants

within the paper and its Supporting Information files.

**Funding:** This study was funded by the Stanley Thomas Johnson Foundation (1053-KF, received by DHP; https://www.johnsonstiftung.ch/), and the U.S. Centers for Disease Control (CDC) who funded the original brucellosis study (received by RT; https://www.cdc.gov/). The funders had no role in study design, data collection and analysis, decision to publish, or preparation of the manuscript.

**Competing interests:** The authors have declared that no competing interests exist.

ongoing integrated human and animal surveillance requiring close collaboration of the human and animal health sectors with community representatives following a One Health approach.

## Author summary

Q fever and Rift Valley fever are two diseases that can affect both humans and animals, causing illness and death. These two diseases can cause large-scale outbreaks, not only affecting the health of individuals and communities but also resulting in a substantial economic loss at the individual, community, regional and national levels through losses in livestock products. We conducted a sero-epidemiological study in the Afar region of north-eastern Ethiopia to determine the presence of these two diseases in nomadic pastoralists and their livestock. Our results indicate that 25.0% of the pastoralists and 34.3% of their animals have had previous exposure to Q fever and 6.1% of the pastoralists and 3.9% of their animals have been exposed to Rift Valley fever. Goats appear to be the most common livestock species associated with Q fever in this region, while cattle are the most common species associated with Rift Valley fever. Our findings highlight the importance of continuous surveillance; additionally, we would recommend the development of rapid response plans for potential outbreaks that integrate the human and animal health sectors.

## Introduction

Zoonotic diseases can directly affect the health of humans, animals and their environment, as well as creating a substantial economic burden at the individual, community, regional, and national levels. Ethiopia is among the top five countries in the world for zoonotic disease burden [1]. However, Q fever and Rift Valley fever (RVF) are two zoonotic diseases that are often under-diagnosed and under-reported due to their non-specific flu-like symptoms, diagnostic difficulties, poor infrastructure, poor public education and engagement, contributing to the burden of undiagnosed febrile illness in this region of Africa [2–4].

Q fever is caused by *Coxiella burnetii*, a zoonotic bacterium found globally with the primary reservoirs for human infection in cattle, sheep and goats [5]. Transmission to humans is predominantly through inhalation of aerosolized bacteria from birth products of infected animals, and direct contact with infected animals or their secretions, raw milk consumption as well as tick-borne transmission [5–7]. Acute Q fever can cause flu-like symptoms, including pneumonia and acute hepatitis [8–10]. Approximately 20% of patients develop Q fever fatigue syndrome (QFS), lasting from six months to many years, with patients suffering prolonged periods of fatigue among other symptoms [11]. Chronic Q fever in humans evolves from 1–5% of acute infections to complications such as: endocarditis, chronic vascular infections, osteomyelitis, osteoarthritis, chronic pulmonary infections, and chronic hepatitis [10]. *C. burnetii* infection is often associated with occupations that have close contact with animals, such as livestock farmers, slaughterhouse workers, butchers, veterinarians, and laboratory workers [8]. Animal infection, while mostly asymptomatic, can cause reproductive disorders in ruminants, including abortion, stillbirth, infertility, mastitis, and endometritis–with significant economic consequence for the livestock industry [8,12].

The RVF virus (RVFV) primarily affects animals (livestock and wildlife) but can infect humans as well. Transmission to humans can occur through mosquitoes and other

hematophagous arthropods (vector-borne disease) or direct contact with infected animal tissues and fluids [13,14]. RVFV infection usually causes a mild non-specific febrile illness in humans, with the potential to progress to a haemorrhagic fever-like syndrome, meningo-encephalitis, or ocular disease. In animals, RVF is more severe with high fatalities. Vertical transmission from mother to fetus occurs in both humans and animals, commonly causing abortion in livestock [13]. A study in Sudan also reported an association between RVFV infection and miscarriage in pregnant women [15].

A scoping review on zoonotic diseases in the Horn of Africa that included 2,055 studies found Q fever to be comparatively understudied across the region, and both Q fever and RVF to be understudied in Ethiopia with each investigated in only 1% of publications [16]. *C. burnetii* and RVFV exposure has previously been reported in livestock in the southern and eastern regions of Ethiopia [17–21]. Information on the prevalence of these two pathogens in humans in Ethiopia is limited, although one study on pastoralists from the Somali region reported a seroprevalence of 27% and 13% for *C. burnetii* and RVFV antibodies, respectively [18].

The Afar region in north-eastern Ethiopia presents an important study region for zoonotic pathogens, with a large pastoralist population and mixture of livestock species. It is a major pastoral region in consideration of livestock numbers and importance to the regional economy in addition to cross-border movements of both humans and livestock to neighbouring countries within the Horn of Africa [22]. Following the recent conflict in the Tigray region of Ethiopia, Afar has also seen an influx of internally displaced people migrating to the region. Pastoralists in this region depend heavily on their livestock for meat and milk, either for their own consumption or to sell. Milk is the primary source of vitamin A for pastoral communities who traditionally have not had access to fruits and vegetables [23]. Animal losses directly affect the health of pastoralists, causing malnutrition and vitamin deficiencies [23]. This makes them particularly vulnerable to environmental changes, such as drought and zoonotic disease [24]. During the dry season people are migrating in search for grazing areas and water with their livestock [25]. This high rate of animal mobility is a potentially important factor in disease transmission. Additionally, the nomadic lifestyle of pastoralists creates a challenge for delivery of health services and disease surveillance [26]. In recent years, a substantial increase of migrants in the Afar region was due to conflicts in Eritrea and the northern Ethiopia region of Tigray. The nomadic pastoralists and increasingly mobile and vulnerable refugee populations, living in low-resource and poor hygienic conditions with high exposure to animals provide a rationale for improved awareness and evidence-based research to support the development of sustainable disease surveillance and control programs to reduce the burden of zoonotic diseases in Ethiopia.

Studies on zoonotic diseases in the Horn of Africa often have not followed a One Health approach, but rather consider human and animal health individually [16]. However, to adequately control zoonotic diseases in these pastoral regions, a transdisciplinary approach is needed, involving the pastoral communities, the animal and public health authorities along with the research community [23].

This study aimed to determine the seroprevalence of *C. burnetii* and RVFV antibodies in humans and their livestock using a One Health approach and to identify associated risk factors in the Afar region of north-eastern Ethiopia. We hypothesized that pastoralists and livestock residing in the Afar region would have high rates of exposure to these pathogens due to the close contact between humans and animals, high rates of animal mobility, and limited medical and veterinary services in this area. The results of this study will be provided to the local health and veterinary authorities to guide in their priorities and practices for controlling zoonoses in Afar.

## Methods

### Ethics statement

The studies were performed in accordance with the principles of the Declaration of Helsinki and were approved by the Armauer Hansen Research Institute and Alert Hospital (AHRI/ ALERT) Ethics Review Committee (PO-53-22) and "Ethikkommission Nordwest- und Zentralschweiz" (EKNZ) (AO_2022–0052).

### Study design and sample collection

This cross-sectional study examined serum samples and epidemiological data (household questionnaire) collected from a recent One Health study on brucellosis [22,25]. Sample and data collection methodology of the brucellosis study are available in Tschopp *et al* (2022) [22]. In addition to informed written consent for the brucellosis study, a general written consent was obtained for further investigation of collected samples for zoonotic diseases, including Q fever and RVF. Serum samples and epidemiological data were collected between 2017 and 2022 from pastoralists and their livestock (sheep, goats, cattle and camels) in seven districts (*woredas*) within the Afar region of Ethiopia. A subset of serum samples was randomly selected from the recent One Health brucellosis study, representing five of the seven districts: Amibara, Awash, Asayta, Mille and Dubti.

### Sample size calculation

A sample size of 323 and 87 pastoralists was calculated for *C. burnetii* and RVFV, respectively, using epitools from https://epitools.ausvet.com.au [27]. The sample size was calculated for a precision of 0.05, a confidence of 95% and estimated seroprevalences of 30% (*C. burnetii*) and 6% (RVFV) in the pastoralists [18]. A total of 335 pastoralists were included in the study.

Sample size for animals was similarly calculated with an estimated seroprevalence between 30% and 60% for *C. burnetii* and 6% and 45% for RVFV [18,19,28,29]. The highest sample size required for each species was: 323 sheep, 385 goats, 340 cattle and 369 camels. To determine the presence of correlations in the seroprevalences of Q fever and RVF for pastoralists and livestock, the livestock were selected to match the same households as the pastoralists. As samples were selected from the previous brucellosis study and there were only limited samples of sufficient volume for inclusion in the present study; consequently we included 684 goats, 199 sheep, 236 cattle and 258 camels.

### Serological testing

Laboratory investigations were performed at the Armauer Hansen Research Institute (AHRI) in Addis Ababa, Ethiopia. For Q fever, serum from pastoralists (n = 335) was tested with the commercially available CE-labeled Fuller Laboratories *Coxiella burnetii* IFA IgG Phase I/II kit (Fuller Laboratories, Fullerton, CA, USA). Serum was screened at a titre of 1:32, with a positivity cut-off titre set at $\geq$1:32 [30,31]. A total of 1,377 sera from livestock were screened using the ID Screen Q fever indirect multi-species ELISA (ID.vet, Innovative Diagnostics, Grabes, France). The $S_{ample}/P_{ositive}$% was calculated for each sample [S/P% = ($OD_{sample}$−mean $OD_{negative\ control}$)/(mean $OD_{positive\ control}$−mean $OD_{negative\ control}$)]. The results were interpreted as negative (S/P% $\leq$ 50%) or positive (S/P% > 50%).

For RVF, serum from pastoralists (n = 335) and their livestock (n = 1,377) were tested with the ID Screen Rift Valley fever competition multi-species ELISA (ID.vet, Innovative Diagnostics, Grabes, France). The S/P% was calculated as above. The results were interpreted as positive (S/P% $\leq$ 40%) or negative (S/P% > 40%).

An independent evaluation of the ID.Screen Rift Valley fever multi-species ELISA determined a diagnostic sensitivity of 85.4% and specificity of 98.6% with the manufacture's cut-off as described above [32]. The sensitivity and specificity of the ID.Screen Q fever multi-species ELISA could not be found in the literature in an independent evaluation, however, the manufacturer reported 100% for both sensitivity and specificity.

## Statistical analysis

Statistical analysis was performed using Stata/IC 16.1. Descriptive statistics and seroprevalence of *C. burnetii* and RVFV antibodies were calculated for the pastoralists, total livestock, and for each species. For *C. burnetii* in humans, seroprevalence was calculated based on composite results of IgG Phase I and/or Phase II positivity.

Uni- and multi-variable logistic regression models with a random effect on village level to account for clustering were performed to identify factors associated with individual level seropositivity. Regression analysis was performed for pastoralists and total livestock. Variables from the household questionnaire relating to contact between pastoralists and livestock, movement of livestock, and symptoms of Q fever and RVF in livestock were included in the regression analysis (Table 1). Variables with a p-value ≤ 0.2 in the univariable analysis were selected for inclusion in the multivariable analysis with the model of best fit determined by the likelihood ratio test. Collinearity between included variables were checked for using the Pearson correlation coefficient. A p-value < 0.05 was considered significant.

## Results

This cross-sectional study included 335 pastoralists from 249 households in 32 villages. Pastoralists were aged between 7 and 80 years with a median age of 35 years (IQR 25–45). Men accounted for 55.2% of included pastoralists. Pastoralists came from five districts: Amibara (n = 109), Awash (n = 60), Asayta (n = 70), Mille (n = 80) and Dubti (n = 16) (Fig 1). Additionally, 1,377 livestock were tested to determine seroprevalence, including: goats (n = 684), sheep (n = 199), cattle (n = 236) and camels (n = 258). The majority of the livestock were female (92.4%) and of breeding age (95.5%). Household questionnaire data matched to both pastoralist and livestock seroprevalence data were available for 239 households in 32 villages. Livestock abortions were reported by 32.6% (78/239) of households spread throughout 32 villages, with 44.9% of these occurring in the late stage of pregnancy (towards the end of gestation) with a further 42.3% reported as mixed, including early (first trimester for large ruminants or first semester for small ruminants) to late stage abortions. Additionally, 35.1% (84/239) of households reported stillbirths among their livestock. All households interviewed in this study reported the practice of animal afterbirth disposal by discarding it in the bush. Selling livestock in the past 12 months was reported by 67.8% of households, while only 22.2% reported purchasing of new animals. The majority of households (72.4%) reported migration with their livestock.

### Seroprevalence of *C. burnetii* and RVFV in pastoralists and livestock

The *C. burnetii* and RVFV seroprevalences in pastoralists were 25.0% (95% CI 18.6–32.6) and 6.1% (95% CI 3.3–11.0), respectively (Table 2). The overall seroprevalence of *C. burnetii* in livestock was 34.3% (95% CI 27.9–41.3) and 3.9% (95% CI 2.6–5.7) for RVFV. Goats had the highest seroprevalence for *C. burnetii* (51.9%, 95% CI 44.9–58.8), while cattle had the highest for RVFV (8.3%, 95% CI 3.3–19.2) (Table 2).

No correlation in seroprevalence of *C. burnetii* or RVFV was observed between humans and livestock within the same household.

**Table 1.  Variables included logistic regression analysis.**

| | Description for pastoralists | Description for livestock |
|---|---|---|
| **Individual level variables** | | |
| Sex | | Female |
| | | Male |
| Age | ≤15 years | Breeder (*camels ≥ 4 years; cattle: ≥ 3 years; sheep/goats ≥ 6 months*) |
| | 16–31 years | Young (*camels < 4 years; cattle: < years; sheep/goats < 6 months*) |
| | 32–48 years | |
| | ≥49 years | |
| District | | Amibara |
| | | Awash |
| | | Asayta |
| | | Mille |
| | | Dubti |
| Species | Not applicable | Camel |
| | | Cattle |
| | | Sheep |
| | | Goat |
| **Household level variables** | | |
| Livestock ownership | Camel | Not applicable |
| | Cattle | |
| | Sheep | |
| | Goat | |
| Camel abortion event in the household in the past 12 months | | No abortion events in the herds |
| | | Abortion events in the herds |
| | | No camels owned |
| Cattle abortion event in the household in the past 12 months | | No abortion events in the herds |
| | | Abortion events in the herds |
| | | No cattle owned |
| Sheep abortion event in the household in the past 12 months | | No abortion events in the herds |
| | | Abortion events in the herds |
| | | No sheep owned |
| Goat abortion event in the household in the past 12 months | | No abortion events in the herds |
| | | Abortion events in the herds |
| | | No goats owned |
| Abortion period | | No abortion events in the herds |
| | | Early (*first semester for small ruminants; first trimester for large ruminants*) |
| | | Late (*second semester for small ruminants; second and third trimester for large ruminants*) |
| | | Mix of both early and late abortions |
| Livestock stillborn in the past 12 months | | No |
| | | Yes |
| Migration in the past 12 months | | No |
| | | Yes |
| Livestock purchased in the past 12 months | | No |
| | | Yes |

(*Continued*)

**Table 1.** (Continued)

|  | Description for pastoralists | Description for livestock |
|---|---|---|
| Livestock sold in the past 12 months |  | No |
|  |  | Yes |
| Men involved in shepherding of livestock |  | No |
|  |  | Yes |
| Women involved in shepherding of livestock |  | No |
|  |  | Yes |
| Children* involved in shepherding of livestock |  | No |
|  |  | Yes |

*As identified by the respondent.

## Risk factors associated with Coxiella burnetii seropositivity in pastoralists

Univariable analysis was performed to identify risk factors associated with *C. burnetii* seropositivity in pastoralists (Table 3). Pastoralists from Asayta (OR = 0.36, 95% CI 0.15–0.87, p = 0.022) and Mille (OR = 0.42, 95% CI 0.18–0.98, p = 0.045) had a lower odds ratio (OR) of

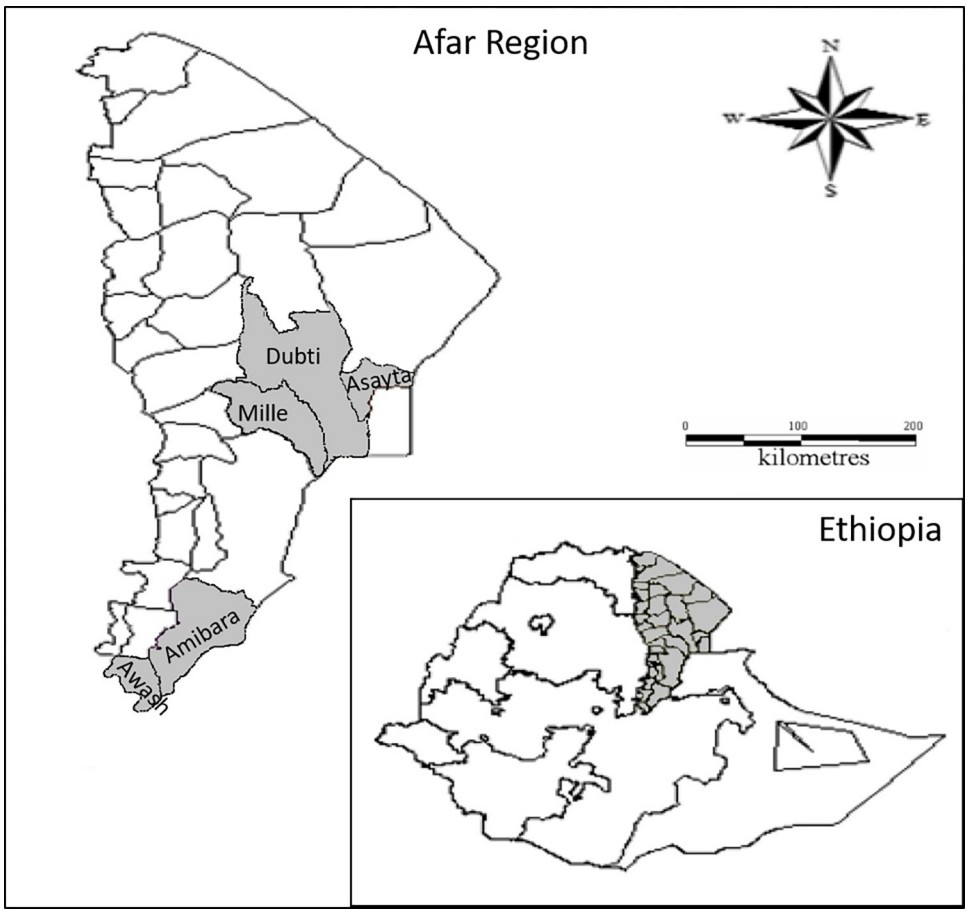

**Fig 1. Map of Afar, north–eastern Ethiopia.** Study sites shown in grey. Insert shows a map of Ethiopia with the Afar region in grey. Adapted from [33].

**Table 2. Seroprevalence of Coxiella burnetii and Rift Valley fever virus in pastoralists and livestock in Afar, Ethiopia.**

| | *C. burnetii* positive | | | RVFV positive | | |
|---|---|---|---|---|---|---|
| | **Positive, n** | **%** | **95% CI** | **Positive, n** | **%** | **95% CI** |
| **Pastoralists** | 91 | 25.0 | 18.6–32.6 | 25 | 6.1 | 3.3–11.0 |
| **Livestock** | 510 | 34.3 | 27.9–41.3 | 63 | 3.9 | 2.6–5.7 |
| Camels | 42 | 16.3 | 10.4–21.6 | 6 | 1.8 | 0.4–6.9 |
| Cattle | 23 | 8.8 | 5.0–15.0 | 32 | 8.3 | 3.3–19.2 |
| Sheep | 85 | 39.9 | 24.6–51.2 | 5 | 2.5 | 1.0–5.9 |
| Goats | 360 | 51.9 | 44.9–58.8 | 20 | 2.7 | 1.4–5.1 |

*C. burnetii* seropositivity than those from Amibara. Pastoralists ≥ 49 years old (OR = 2.82, 95% CI 1.34–5.93, p = 0.006) and children ≤ 15 years old (OR = 3.83, 95% CI 1.35–10.81, p = 0.011) had an increase in OR for being seropositive for *C. burnetii* (OR = 2.82, 95% CI 1.34–5.93, p = 0.006) than adults aged between 16 and 31 years of age. Further, pastoralists from households where children were involved in shepherding of livestock had an increased OR of being seropositive for *C. burnetii* (OR = 2.24 95% CI 1.16–4.35, p = 0.017). Pastoralists from households reporting livestock abortions, across both early and late pregnancy periods, had an increased OR of *C. burnetii* seropositivity, compared to no abortions in livestock (OR = 3.16, 95% CI 1.43–7.00, p = 0.004). Pastoralists from households that reported abortion events in goats had a 2.1-fold (95% CI 1.18–3.78, p = 0.011) increase in OR of *C. burnetii* infection compared to households that reported no goat abortions.

The multivariable model of best fit for *C. burnetii* seropositivity in pastoralists included age and a history of goat abortions in the household. Pastoralists aged ≤ 15 years (OR = 4.29, 95% CI 1.47–12.54, p = 0.008) and those aged ≥ 49 (OR = 2.31, 95% CI 1.06–5.05, p = 0.036) had an increased OR for seropositivity to *C. burnetii*. Pastoralists from households spread throughout 32 villages, that reported abortion events in goats had an increased OR of *C. burnetii* seropositivity (OR = 2.13, 95% CI 1.17–3.86, p = 0.013).

## Risk factors associated with Rift Valley fever virus seropositivity in pastoralists

Univariable analysis was performed to identify risk factors associated with RVFV seropositivity in pastoralists (Table 4). Pastoralists ≥ 49 years old had an increase in OR for being seropositive for RVFV (OR = 3.29, 95% CI 1.02–10.61, p = 0.036) than adults aged between 16 and 31 years of age.

No other factors were associated with RVF seropositivity, thus multivariable analysis was not done.

## Risk factors associated with *Coxiella burnetii* seropositivity in livestock

Univariable analysis was performed to identify risk factors associated with *C. burnetii* seroprevalence in livestock (Table 5). Sheep (OR = 0.56, 95% CI 0.40–0.80, p = 0.001), cattle (OR = 0.09, 95% CI 0.05–0.14, p < 0.001) and camels (OR = 0.16, 95% CI 0.10–0.24, p < 0.001) had a significantly lower OR of being seropositive for *C. burnetii* compared to goats. Young animals also had a lower OR (OR = 0.07, 95% CI 0.02–0.30, p < 0.001) of being *C. burnetii* seropositive compared to those of breeding age. Livestock in Mille had decreased OR (OR = 0.42, 95% CI 0.21–0.84, p = 0.014) of being seropositive for *C. burnetii* compared to those in Amibara. Livestock from households that did not have goats had a lower OR of *C. burnetii* seropositivity than those from households that had goats, even without abortion events.

**Table 3. Univariable and multivariable analysis of predictors for *Coxiella burnetii* seropositivity in pastoralists.**

| | Positive (%) | Univariable | | | Multivariable | | |
|---|---|---|---|---|---|---|---|
| | | OR | 95% CI | P-value | AOR | 95% CI | P-value |
| *Individual variables* | | | | | | | |
| **Sex** | | | | | | | |
| Male | 47/185 (25.4) | 1.00 | | | | | |
| Female | 45/150 (30.0) | 1.22 | 0.73–2.05 | 0.441 | | | |
| **Age** | | | | | | | |
| ≤15 | 10/22 (45.5) | 3.83 | 1.35–10.81 | *0.011* | 4.29 | 1.47–12.54 | *0.008* |
| 16–31 | 27/129 (20.9) | 1.00 | | | 1.00 | | |
| 32–48 | 32/126 (25.4) | 1.36 | 0.73–2.55 | 0.337 | 1.24 | 0.65–2.38 | 0.512 |
| ≥49 | 23/58 (39.7) | 2.82 | 1.34–5.93 | *0.006* | 2.31 | 1.06–5.05 | *0.036* |
| **District** | | | | | | | |
| Amibara | 41/109 (37.6) | 1.00 | | | | | |
| Awash | 17/60 (28.3) | 0.64 | 0.23–1.58 | 0.330 | | | |
| Asayta | 13/70 (18.6) | 0.36 | 0.15–0.87 | *0.022* | | | |
| Mille | 16/80 (20.0) | 0.42 | 0.18–0.98 | *0.045* | | | |
| Dubti | 5/16 (31.3) | 0.78 | 0.20–3.07 | 0.727 | | | |
| *Household variables* | | | | | | | |
| **Livestock ownership** | | | | | | | |
| Camel | 66/246 (26.8) | 0.83 | 0.39–1.75 | 0.621 | | | |
| Cattle | 68/255 (26.7) | 0.81 | 0.37–1.75 | 0.585 | | | |
| Sheep | 59/214 (27.6) | 1.03 | 0.57–1.89 | 0.914 | | | |
| Goat | 87/310 (28.1) | 5.15 | 0.61–43.77 | 0.133 | | | |
| **Camel abortion event in the household in the past 12 months** | | | | | | | |
| No abortion | 61/231 (26.4) | 1.00 | | | | | |
| Abortion | 5/15 (33.3) | 1.93 | 0.56–6.57 | 0.295 | | | |
| No camels | 22/78 (28.2) | 1.27 | 0.59–2.73 | 0.545 | | | |
| **Cattle abortion event in the household in the past 12 months** | | | | | | | |
| No abortion | 63/236 (26.7) | 1.00 | | | | | |
| Abortion | 5/19 (26.3) | 1.69 | 0.46–6.24 | 0.431 | | | |
| No cattle | 20/69 (29.0) | 1.31 | 0.59–2.95 | 0.506 | | | |
| **Sheep abortion event in the household in the past 12 months** | | | | | | | |
| No abortion | 59/211 (28.0) | 1.00 | | | | | |
| Abortion | 0/4 (0.0) | Omitted | | | | | |
| No sheep | 29/109 (26.6) | 0.93 | 0.50–1.70 | 0.804 | | | |
| **Goat abortion event in the household in the past 12 months** | | | | | | | |
| No abortion | 53/222 (23.9) | 1.00 | | | | | |
| Abortion | 34/88 (28.6) | 2.11 | 1.18–3.78 | *0.011* | 2.13 | 1.17–3.86 | *0.013* |
| No goats | 1/14 (7.1) | 0.23 | 0.03–0.42 | 0.186 | 0.21 | 0.02–1.93 | 0.169 |
| **Abortion period** | | | | | | | |
| No abortion | 53/226 (23.5) | 1.00 | | | | | |
| Early | 1/4 (25.0) | 1.94 | 0.14–26.54 | 0.617 | | | |
| Late | 14/44 (31.8) | 1.56 | 0.73–3.37 | 0.254 | | | |
| Mix | 18/41 (43.9) | 3.16 | 1.43–7.00 | *0.004* | | | |
| Not answered | 2/9 (22.2) | 0.85 | 0.15–4.81 | 0.854 | | | |
| **Livestock stillborn in the past 12 months** | 35/120 (29.2) | 1.10 | 0.63–1.92 | 0.726 | | | |
| **Migration in the past 12 months** | 62/231 (26.8) | 0.82 | 0.41–1.66 | 0.575 | | | |
| **Livestock purchased in the past 12 months** | 17/67 (25.4) | 0.84 | 0.43–1.63 | 0.601 | | | |

*(Continued)*

**Table 3.** (Continued)

| | Positive (%) | Univariable | | | Multivariable | | |
|---|---|---|---|---|---|---|---|
| | | OR | 95% CI | P-value | AOR | 95% CI | P-value |
| Livestock sold in the past 12 months | 61/221 (27.6) | 1.17 | 0.62–2.21 | 0.635 | | | |
| Men involved in shepherding of livestock | 23/105 (21.9) | 0.63 | 0.34–1.16 | 0.137 | | | |
| Women involved in shepherding of livestock | 65/212 (30.7) | 1.73 | 0.91–3.27 | 0.094 | | | |
| Children involved in shepherding of livestock | 70/224 (31.3) | 2.24 | 1.16–4.35 | ***0.017*** | | | |

Statistically significant (p–value ≤ 0.05) variables are highlighted in bold italics. OR = Odds ratio. AOR = Adjusted odds ratio.

In the multivariable analysis the model of best fit for *C. burnetii* seropositivity in livestock included district, sex, age, species, abortion in goats and shepherding by women. Sheep (OR = 0.60, 95% CI 0.42–0.86, p = 0.005), cattle (OR = 0.08, 95% CI 0.05–0.14, p < 0.001) and camels (OR = 0.18, 95% CI 0.11–0.27, p < 0.001) still showed a reduced OR for *C. burnetii* seropositivity compared with goats. In the multivariable analysis, Asayta (OR = 0.48, 95% CI 0.29–0.80, p = 0.005) also showed reduced odds of *C. burnetii* seropositivity in individual animals, along with Mille (OR = 0.31, 95% CI 0.18–0.52, p < 0.001). Livestock in households where women were involved in shepherding also had a lower OR of *C. burnetii* seropositivity (OR = 0.67, 95% CI 0.47–0.94, p = 0.020).

## Risk factors associated with Rift Valley fever virus seropositivity in livestock

Univariable analysis was performed to identify risk factors associated with RVFV seroprevalence in livestock (Table 6). Livestock in Dubti had a 7.0-fold (95% CI 2.58–18.75, p < 0.001) increase in OR of being seropositive for RVFV compared to those in Amibara. Cattle had a 6.6-fold (95% CI 3.29–13.31, p < 0.001) increase in OR of being seropositive to RVFV compared to goats, with livestock from households that reported abortion events in cattle having a 2.5-fold (95% CI 10.5–6.08, p = 0.039) increase in odds. Livestock from households reporting early-term (OR = 7.97, 95% CI 1.49–42.62, p = 0.015) abortions in livestock had an increased OR of RVFV seropositivity compared to those that did not report abortion events in livestock.

In the multivariable analysis, the model of best fit for RVFV seropositivity in livestock included district, species, abortion events in cattle, abortion periods and shepherding by women. The increase in OR of seropositivity was similar to that of the univariable analysis with cattle having a higher OR (OR = 4.35, 95% CI 1.82–10.36, p = 0.001) compared with goats and animals in Dubti having a higher OR (OR = 10.62, 95% CI 3.16–35.72, p < 0.001) compared with those in Amibara.

## Discussion

In our study we employed a One Health approach to investigate the relationship between seropositivity of Q fever and RVFV in pastoralists and their livestock in Afar, north-eastern Ethiopia. We found a quarter (25.0%) of the pastoralists were seropositive for Q fever, matching a similar study by Ibrahim *et al.* from the Somali region of Ethiopia that found 27% seroprevalence to *C. burnetii* in pastoralists [18]. These results, however, are in contrast to the only other study to date in humans in Ethiopia, in which Addis Ababa abattoir workers had a 6.5% prevalence for Q fever [34]. While abattoir workers are considered a high-risk group [8], these results may indicate that pastoralists are at even higher risk, possibly due to life-long close contact with their livestock representative of a cumulative risk. A recent study in Chad that looked

**Table 4. Univariable analysis of predictors for Rift Valley fever virus seropositivity in pastoralists.**

| | Positive (%) | OR | 95% CI | P-value |
|---|---|---|---|---|
| *Individual variables* | | | | |
| **Sex** | | | | |
| Male | 16/185 (8.6) | 1.00 | | |
| Female | 9/150 (6.0) | 0.69 | 0.28–1.68 | 0.411 |
| **Age** | | | | |
| ≤15 | 0/22 (0.0) | Omitted | | |
| 16–31 | 6/129 (4.7) | 1.00 | | |
| 32–48 | 11/126 (8.7) | 1.86 | 0.64–5.42 | 0.253 |
| ≥49 | 8/58 (13.8) | 3.29 | 1.02–10.61 | ***0.046*** |
| **District** | | | | |
| Amibara | 5/109 (4.7) | 1.00 | | |
| Awash | 5/60 (8.3) | 2.03 | 0.43–9.54 | 0.368 |
| Asayta | 10/70 (14.3) | 3.69 | 0.99–13.80 | 0.052 |
| Mille | 5/80 (6.3) | 1.59 | 0.36–7.05 | 0.539 |
| Dubti | 0/16 (0.0) | Omitted | | |
| *Household variables* | | | | |
| **Livestock ownership** | | | | |
| Camel | 16/246 (6.5) | 0.94 | 0.27–3.26 | 0.919 |
| Cattle | 17/255 (6.7) | 0.97 | 0.27–3.51 | 0.966 |
| Sheep | 17/214 (7.9) | 2.10 | 0.69–6.43 | 0.193 |
| Goat | 23/310 (7.4) | Omitted | | |
| **Camel abortion event in the household in the past 12 months** | | | | |
| No abortion | 15/231 (6.5) | 1.00 | | |
| Abortion | 1/14 (7.1) | 0.86 | 0.09–8.05 | 0.893 |
| No camels | 7/78 (9.0) | 1.05 | 0.30–3.73 | 0.935 |
| **Cattle abortion event in the household in the past 12 months** | | | | |
| No abortion | 15/236 (6.4) | 1.00 | | |
| Abortion | 2/19 (10.5) | 1.31 | 0.22–7.77 | 0.765 |
| No cattle | 6/69 (8.7) | 1.07 | 0.29–3.92 | 0.921 |
| **Sheep abortion event in the household in the past 12 months** | | | | |
| No abortion | 17/211 (8.1) | 1.00 | | |
| Abortion | 0/4 (0.0) | Omitted | | |
| No sheep | 6/109 (5.5) | 0.49 | 0.16–1.49 | 0.208 |
| **Goat abortion event in the household in the past 12 months** | | | | |
| No abortion | 19/222 (8.6) | 1.00 | | |
| Abortion | 4/88 (4.5) | 0.53 | | |
| No goats | 0/14 (0.0) | Omitted | 0.16–1.72 | 0.292 |
| **Abortion period** | | | | |
| No abortion | 18/226 (6.8) | 1.00 | | |
| Early | 0/4 (0.0) | Omitted | | |
| Late | 2/44 (4.5) | 0.61 | 0.12–2.95 | 0.534 |
| Mix | 3/41 (7.3) | 0.88 | 0.22–3.46 | 0.852 |
| Not answered | 0/9 (0.0) | Omitted | | |
| **Livestock stillborn in the past 12 months** | 7/120 (5.8) | 0.73 | 0.27–1.96 | 0.536 |
| **Migration in the past 12 months** | 15/231 (6.5) | 0.88 | 0.29–2.71 | 0.824 |
| **Livestock purchased in the past 12 months** | 5/67 (7.5) | 0.99 | 0.33–3.02 | 0.990 |
| **Livestock sold in the past 12 months** | 19/221 (8.6) | 2.66 | 0.79–9.00 | 0.116 |

*(Continued)*

**Table 4.** (Continued)

| | Positive (%) | OR | 95% CI | P-value |
|---|---|---|---|---|
| **Men involved in shepherding of livestock** | 9/105 (5.6) | 1.19 | 0.45–3.16 | 0.723 |
| **Women involved in shepherding of livestock** | 11/212 (5.2) | 0.52 | 0.19–1.41 | 0.199 |
| **Children involved in shepherding of livestock** | 13/224 (5.8) | 0.85 | 0.47–2.64 | 0.781 |

Statistically significant (p–value ≤ 0.05) variables are highlighted in bold italics. OR = Odds ratio.

at 960 mobile agro-pastoralists found an even higher prevalence of 49.7% for *C. burnetii* antibodies [35].

A third of the livestock (34.3%) were seropositive for *C. burnetii*, with goats having the highest AP (51.9%), followed by sheep (36.9%), camels (16.3%) and cattle (8.8%). The differences in seroprevalence between species was suggested by a study by Tschopp *et al.* to be the result of pastoralists in Afar often keeping herds separated by species [25]. Goats are the most commonly sold species, potentially exposing them to other infected animals at markets [25]. Regular market exposure highlights the potential risk of infected goats transmitting the disease to new (uninfected/unexposed) herds or unsold animals returning to their original herds with newly acquired pathogens.

In our study, we found an association between children ≤ 15 years and *C. burnetii* seropositivity. Children in the Afar region often assist in caring for goats in addition to regularly being fed raw milk [25]. A study on risk factors of zoonoses among pastoralists in Afar found all participants drank raw milk with goat milk being the most commonly consumed and 20% of participants also reported consuming soured milk [25]. Further, pastoralists from households that reported abortion events in goats had a 2.1-fold increase in OR of being seropositive compared to pastoralists from households without goat abortions. These findings suggest goats may be a primary cause of *C. burnetii* transmission to humans in this region.

The study in the Somali region of Ethiopia reported similar seroprevalence results for goats (48.8%), sheep (28.9%) and cattle (9.6%), however, reported a much higher seroprevalence in camels (55.7%) [18]. A study in livestock from pastoral regions in south-eastern Ethiopia also found a similar seroprevalence in goats (54.2%), but higher in camels (90.0%) and cattle (31.6%) [17]. While a study in small ruminants from the Borana pastoral area in southern Ethiopia found lower seroprevalences with 35.7% in goats and 18.3% in sheep [19]. The study in Chad also found lower seroprevalences of 7.1% in cattle, 17.1% in goats and 19.1% in sheep [35]. Although direct comparisons cannot be made due to differences in testing methods, the variation in reported seroprevalences could be attributed to differences in animal husbandry methods or environmental conditions including the presence of ticks. Pastoralists from Asayta and Mille in central Afar had significantly lower OR of being *C. burnetii* seropositive than those from Amibara in southern Afar, bordering the Somali and Oromia regions of Ethiopia. Similarly, in the multivariable analysis livestock from Asayta and Mille had a significantly decreased OR of *C. burnetii* compared to livestock in Amibara.

While no association was seen with sex for either humans or their livestock, the majority of animals included were female due to pastoralists primarily keeping animals for milk production and reproduction purposes [18]. A positive association was observed for pastoralists aged ≥ 49 years with *C. burnetii* seropositivity, this is often observed due to an individual having more cumulative time to become exposed to the pathogen [18]. Similarly, in livestock, young animals also had a significantly lower OR of being *C. burnetii* seropositive compared with animals of breeding age.

**Table 5. Univariable and multivariable analysis of predictors for *Coxiella burnetii* seropositivity in livestock.**

| | | Univariable | | | Multivariable | | |
|---|---|---|---|---|---|---|---|
| | Positive (%) | OR | 95% CI | P-value | AOR | 95% CI | P-value |
| *Individual variables* | | | | | | | |
| **Sex** | | | | | | | |
| Female | 479/1272 (37.7) | 1.00 | | | 1.00 | | |
| Male | 31/105 (29.5) | 0.60 | 0.38–0.96 | | 0.71 | 0.41–1.20 | 0.197 |
| **Age** | | | | | | | |
| Breeder | 508/1315 (38.6) | 1.00 | | | 1.00 | | |
| Young | 2/60 (3.3) | 0.07 | 0.02–0.30 | *<0.001* | 0.23 | 0.05–1.02 | 0.053 |
| Unknown | 0/2 (0.0) | Omitted | | | | | |
| **Species** | | | | | | | |
| Goat | 360/684 (52.6) | 1.00 | | | 1.00 | | |
| Sheep | 85/199 (42.7) | 0.56 | 0.40–0.80 | *0.001* | 0.60 | 0.42–0.86 | *0.005* |
| Cattle | 23/236 (10.6) | 0.09 | 0.05–0.14 | *<0.001* | 0.08 | 0.05–0.14 | *<0.001* |
| Camel | 42/258 (16.3) | 0.16 | 0.10–0.24 | *<0.001* | 0.18 | 0.11–0.27 | *<0.001* |
| **District** | | | | | | | |
| Amibara | 199/560 (35.5) | 1.00 | | | 1.00 | | |
| Awash | 119/224 (53.1) | 1.46 | 0.65–3.30 | 0.361 | 0.71 | 0.40–1.27 | *0.250* |
| Asayta | 93/214 (43.5) | 1.36 | 0.71–2.60 | 0.351 | 0.48 | 0.29–0.80 | *0.005* |
| Mille | 71/310 (22.9) | 0.42 | 0.21–0.84 | *0.014* | 0.31 | 0.18–0.52 | *<0.001* |
| Dubti | 28/69 (40.6) | 1.14 | 0.40–3.24 | 0.806 | 1.12 | 0.53–2.38 | 0.765 |
| *Household variables* | | | | | | | |
| **Camel abortion event in the household in the past 12 months** | | | | | | | |
| No abortion | 334/978 (34.2) | 1.00 | | | | | |
| Abortion | 34/67 (50.7) | 1.17 | 0.66–2.07 | 0.591 | | | |
| No camels | 127/281 (45.2) | 0.95 | 0.61–1.46 | 0.798 | | | |
| **Cattle abortion event in the household in the past 12 months** | | | | | | | |
| No abortion | 349/1006 (34.7) | 1.00 | | | | | |
| Abortion | 22/77 (28.6) | 0.84 | 0.44–1.58 | 0.584 | | | |
| No cattle | 124/243 (51.0) | 1.20 | 0.76–1.89 | 0.443 | | | |
| **Sheep abortion event in the household in the past 12 months** | | | | | | | |
| No abortion | 353/990 (35.7) | 1.00 | | | | | |
| Abortion | 2/7 (28.6) | 0.49 | 0.09–2.83 | 0.427 | | | |
| No sheep | 140/329 (42.6) | 1.24 | 0.89–1.72 | 0.199 | | | |
| **Goat abortion event in the household in the past 12 months** | | | | | | | |
| No abortion | 323/877 (36.8) | 1.00 | | | 1.00 | | |
| Abortion | 165/394 (41.9) | 1.11 | 0.85–1.46 | 0.448 | 1.22 | 0.84–1.49 | 0.453 |
| No goats | 7/55 (12.7) | 0.37 | 0.15–0.93 | *0.034* | 0.45 | 0.18–1.10 | 0.080 |
| **Abortion period** | | | | | | | |
| No abortion | 194/483 (40.2) | 1.00 | | | | | |
| Early | 0/14 (0.0) | Omitted | | | | | |
| Late | 78/199 (39.2) | 0.83 | 0.56–1.23 | 0.357 | | | |
| Mix | 80/181 (44.2) | 1.11 | 0.74–1.65 | 0.611 | | | |
| Not answered | 13/39 (33.3) | 0.91 | 0.42–1.96 | 0.813 | | | |
| **Livestock stillborn in the past 12 months** | 180/438 (41.1) | 1.07 | 0.81–1.38 | 0.690 | | | |
| **Migration in the past 12 months** | 341/988 (34.5) | 0.97 | 0.65–1.44 | 0.883 | | | |
| **Livestock purchased in the past 12 months** | 124/315 (39.4) | 1.04 | 0.77–1.39 | 0.812 | | | |
| **Livestock sold in the past 12 months** | 362/583 (62.19 | 1.32 | 0.94–1.84 | 0.108 | | | |

(*Continued*)

**Table 5.** (Continued)

| | Univariable | | | | Multivariable | | |
|---|---|---|---|---|---|---|---|
| | Positive (%) | OR | 95% CI | P-value | AOR | 95% CI | P-value |
| Men involved in shepherding of livestock | 150/384 (39.1) | 1.33 | 0.99–1.80 | 0.059 | | | |
| Women involved in shepherding of livestock | 321/922 (34.8) | 0.74 | 0.53–1.04 | 0.084 | 0.67 | 0.47–0.94 | ***0.020*** |
| Children involved in shepherding of livestock | 361/970 (37.2) | 1.06 | 0.77–1.46 | 0.734 | | | |

Statistically significant (p–value ≤ 0.05) variables are highlighted in bold italics. OR = Odds ratio. AOR = Adjusted odds ratio.

In our study we found a prevalence of 6.1% for RVFV in pastoralists and as with *C. burnetii* those aged ≥ 49 years had an increased OR of seropositivity. The study by Ibrahim *et al.* was the only study identified that reported on RVFV seroprevalence in humans in Ethiopia, this study from the Somali region found a higher prevalence of 13.2% [18]. Further, the study in Chad found a much higher seroprevalence of 28.1% for RVFV among mobile agro-pastoralists [35]. In our study, we found an overall prevalence of RVFV in livestock of 3.9%. Cattle had the highest seroprevalence (8.3%), with the other livestock species being significantly lower: goats (2.7%), sheep (2.5%) and camels (1.8%).

Asebe *et al.* reported a significant association between RVFV infection in cattle and a history of abortion [18]. We saw a similar significant association in the univariable analysis with livestock from households that had a history of abortions in cattle having a 2.5-fold increase in OR of being seropositive for RVFV antibodies. This association, while no longer significant in the multivariable analysis, may be worth further exploration. Moreover, the study investigating brucellosis in Afar found that while there was an association between abortion history and *Brucella* seropositivity in camels, goats and sheep, this association was not present for cattle and postulated there should be a different cause for abortion in cattle [22]. Our study indicates that RVFV could be a contributing agent of abortion in this species.

In contrast to our study, the study in the Somali region reported camels having the highest apparent seroprevalence for RVFV of 42.6%, followed by cattle (17.9%), sheep (7.4%) and goats (6.3%) [18]. Two further studies on RVFV in Ethiopian livestock, which only investigated cattle, reported lower seroprevalences of 5.0% in the south Omo area of southern Ethiopia and 7.6% in south-western Ethiopia, bordering South Sudan [20,21]. While the study in Chad found 9.5% of cattle, 3.9% of goats, and 15.5% of sheep to be seropositive for RVFV [35].

In our study, livestock in Dubti had a 10.6-fold increase in OR of RVFV seropositivity compared to their counterparts in Amibara. The variation in prevalence of RVFV seen in different pastoral areas of Ethiopia and other pastoral communities in Africa may be related to environmental conditions including water sources that directly affect the presence of the mosquito vectors. Dubti neighbours the district of Afambo, in which lie a series of lakes marking the end of the Awash River. These lakes may offer areas of stagnant water, providing breeding grounds for mosquitos, and may be the source of the high seroprevalences of RVFV found in central Afar. A study in southern Ethiopia found *Aedes* mosquitos accounted for 25% of those collected, however further entomological studies would be required to map the density of RVFV vectors in other regions of Ethiopia including Afar [36]. Environmental monitoring of heavy rainfall, flooding, and mosquito swarms have been suggested to identify high risk areas for RVFV control in the Horn of Africa, allowing for targeted livestock vaccination campaigns [37]. This monitoring would provide early warnings for outbreak preparedness programs that may include expanding vaccination coverage to surrounding areas, restrictions on livestock movements or vector control, for example: applying insecticides to livestock [37].

**Table 6. Univariable and multivariable analysis of predictors for Rift Valley fever virus seropositivity in livestock.**

| | Univariable | | | | Multivariable | | |
|---|---|---|---|---|---|---|---|
| | Positive (%) | OR | 95% CI | P-value | AOR | 95% CI | P-value |
| *Individual variables* | | | | | | | |
| **Sex** | | | | | | | |
| Female | 7/105 | 1.00 | | | | | |
| Male | 56/1272 | 1.58 | 0.68–3.66 | 0.285 | | | |
| **Age** | | | | | | | |
| Breeder | 59/1315 (4.5) | 1.00 | | | | | |
| Young | 4/60 (6.7) | 1.61 | 0.54–4.83 | 0.393 | | | |
| Unknown | 0/2 (0.0) | Omitted | | | | | |
| **Species** | | | | | | | |
| Goat | 20/684 (2.9) | 1.00 | | | 1.00 | | |
| Sheep | 5/199 (2.5) | 0.96 | 0.35–2.65 | 0.935 | 1.26 | 0.43–3.71 | 0.678 |
| Cattle | 32/236 (13.6) | 6.62 | 3.29–13.31 | *<0.001* | 4.35 | 1.82–10.36 | *0.001* |
| Camel | 6/258 (2.3) | 0.96 | 0.35–2.60 | 0.930 | 0.51 | 0.13–1.98 | 0.329 |
| **District** | | | | | | | |
| Amibara | 21/560 (3.8) | 1.00 | | | 1.00 | | |
| Awash | 8/224 (3.6) | 0.92 | 0.34–2.48 | 0.862 | 0.91 | 0.23–3.57 | 0.887 |
| Asayta | 8/214 (3.7) | 1.02 | 0.40–2.57 | 0.969 | 0.52 | 0.12–2.24 | 0.378 |
| Mille | 12/310 (3.9) | 1.12 | 0.48–2.64 | 0.792 | 0.56 | 0.12–2.71 | 0.471 |
| Dubti | 14/69 (20.3) | 6.95 | 2.58–18.75 | *<0.001* | 10.62 | 3.16–35.72 | *<0.001* |
| *Household variables* | | | | | | | |
| **Camel abortion event in the household in the past 12 months** | | | | | | | |
| No abortion | 44/978 (4.5) | 1.00 | | | | | |
| Abortion | 6/67 (9.0) | 2.15 | 0.77–6.01 | 0.144 | | | |
| No camels | 11/281 (3.9) | 1.02 | 0.44–2.39 | 0.963 | | | |
| **Cattle abortion event in the household in the past 12 months** | | | | | | | |
| No abortion | 45/1006 (58.4) | 1.00 | | | 1.00 | | |
| Abortion | 9/77 (11.7) | 2.52 | 1.05–6.08 | *0.039* | 2.70 | 0.75–9.73 | 0.129 |
| No cattle | 7/243 (2.9) | 0.67 | 0.26–1.75 | 0.416 | 1.17 | 0.34–4.03 | 0.805 |
| **Sheep abortion event in the household in the past 12 months** | | | | | | | |
| No abortion | 42/990 (4.2) | 1.00 | | | | | |
| Abortion | 0/7 (0.0) | Omitted | | | | | |
| No sheep | 19/329 (5.8) | 1.69 | 0.87–3.28 | 0.118 | | | |
| **Goat abortion event in the household in the past 12 months** | | | | | | | |
| No abortion | 36/877 (4.1) | 1.00 | | | | | |
| Abortion | 22/394 (5.6) | 1.19 | 0.67–2.14 | 0.553 | | | |
| No goats | 3/55 (5.5) | 1.37 | 0.36–5.24 | 0.645 | | | |
| **Abortion period** | | | | | | | |
| No abortion | 14/483 (2.9) | 1.00 | | | 1.00 | | |
| Early | 3/14 (2.1) | 7.97 | 1.49–42.62 | *0.015* | 2.29 | 0.29–17.98 | 0.430 |
| Late | 11/199 (5.5) | 1.71 | 0.71–4.11 | 0.230 | 1.37 | 0.52–3.62 | 0.528 |
| Mix | 12/169 (7.1) | 2.13 | 0.88–5.16 | 0.093 | 1.25 | 0.45–3.47 | 0.671 |
| Not answered | 0/39 (0.0) | Omitted | | | Omitted | | |
| **Livestock stillborn in the past 12 months** | 24/438 (5.5) | 1.10 | 0.62–1.95 | 0.756 | | | |
| **Migration in the past 12 months** | 38/988 (3.8) | 0.75 | 0.36–1.57 | 0.444 | | | |
| **Livestock purchased in the past 12 months** | 12/315 (3.8) | 0.97 | 0.49–1.93 | 0.935 | | | |
| **Livestock sold in the past 12 months** | 40/583 (6.9) | 1.00 | 0.51–1.95 | 0.997 | | | |

*(Continued)*

**Table 6.** (Continued)

| | Univariable | | | | Multivariable | | |
|---|---|---|---|---|---|---|---|
| | Positive (%) | OR | 95% CI | P-value | AOR | 95% CI | P-value |
| **Men involved in shepherding of livestock** | 16/384 (4.2) | 0.78 | 0.41–1.49 | 0.455 | | | |
| **Women involved in shepherding of livestock** | 38/922 (4.1) | 0.58 | 0.29–1.14 | 0.115 | 0.43 | 0.01–0.15 | 0.104 |
| **Children involved in shepherding of livestock** | 48/970 (4.9) | 1.40 | 0.69–2.84 | 0.347 | | | |

Statistically significant (p–value ≤ 0.05) variables are highlighted in bold italics. OR = Odds ratio. AOR = Adjusted odds ratio.

Additional factors influencing RVF prevalence may involve the movement of animals, potentially exposing them to vectors or infected animals from other regions. Large animals, cattle and camels, are generally taken during seasonal migrations, while small ruminants often remain at the settlement, which may account for the higher RVFV antibody prevalence seen in cattle in our study [25]. Pastoralists in different districts use distinct migration pathways, potentially also explaining why RVFV appears to be circulating predominantly in Central Afar while *C. burnetii* was more common in the southern part [26].

No correlation was observed for either *C. burnetii* or RVFV seropositivity between pastoralists and livestock from the same household. It is possible that pastoralists became infected from livestock no longer in their herds, either from selling or death, considering the shorter lifespan of livestock in comparison to humans [22]. Alternatively, pastoralists may have become infected from contact with other animals or their excretions into the environment [22]. *C. burnetii* has been shown to survive in the environment for at least a year [38]. While RVFV could be transmitted by mosquitos from neighbouring herds. A similar study on the seroprevalence of brucellosis in Kyrgyzstan demonstrated clear dependence of human seropositivity from sheep seroprevalence and not from goats or cattle [39]. The study sites included in the Kyrgyzstan study, however, were located at a greater distance from each other than the study sites included in our study in Afar, indicating that correlation of human-animal exposure to zoonoses may be dependent on the scale of comparison. It seems that comparisons over more than a hundred kilometres can be more easily demonstrated because of the higher variability of human-animal contacts at shorter distances [39].

Pastoral regions present a challenge for disease surveillance as well as providing human and veterinary health services, generally being in remote, harsh areas that lack infrastructure including diagnostic facilities. Maintaining the cold chain to these regions for test kits or alternatively to bring samples to a central reference laboratory is often logistically difficult [22]. Correctly diagnosing both humans and animals is essential to ensure appropriate treatment is utilized and prevent antimicrobial resistance increasing in these regions. Current recommended diagnostic tests for Q fever are IFA and for RVF either PCR, virus isolation, or antigen-detection ELISA, all of which are technically demanding and require sophisticated laboratories, which are often found only in urban centres [5,40]. The development of highly sensitive and specific rapid diagnostic tests (RDTs), that are stable at room temperature, would prove particularly beneficial in these pastoral settings, providing immediate results. Additionally, RDTs have been found to be well received by participants who refused venous blood draws [22]. Appropriate diagnostics are also essential for surveillance of these diseases for early detection of potential outbreaks. Large outbreaks in livestock can also see large-scale transmission to humans, as was the case of the Q fever outbreak in the Netherlands from 2007–2010, with more than 4,000 human cases reported and more than 40,000 cases estimated in total [5,41]. The Netherlands outbreak was attributed primarily to dairy goat farming [42].

Furthermore, large outbreaks of zoonotic diseases can cause economic loss at the individual, community and national level. Estimated economic impacts of RVF outbreaks between 1930 and 2009 have ranged from 5–470 million USD at national levels [24]. Ethiopia is particularly vulnerable to large outbreaks of zoonotic disease, having the largest livestock population in Africa [43]. Livestock accounts for 16.5% and 35.6% of the national Gross Domestic Product (GDP) and the agricultural GDP, respectively, supporting 80% of rural inhabitants [43–45]. Since both Q fever and RVF commonly affect domestic ruminants, pastoral livelihoods can be expected to be significantly adversely affected by disease outbreaks.

A limitation of this study is that we made use of samples collected for a previous study and not all samples had enough volume remaining for the Q fever and RVF serological testing. This is evident in only five of the seven districts included in the brucellosis study being included in the present study and also the lower sample numbers for Dubti. Further, only limited samples from sheep, cattle and camels were available and did not meet the required sample size, subsequently regression analysis for individual species seropositivity was not performed. The sample size calculation was done for each pathogen as a single proportion, however, a sample size calculation adjusting for clustering would have been more appropriate. This may have played a role in the broad confidence intervals observed, however, this does not impact our conclusions. An additional limitation of the study was that there was no availability of an independent evaluation of the Q fever ELISA to provide robust diagnostic accuracy data. Serological assays should be evaluated in the population of interest to determine the diagnostic accuracy and most appropriate cut off for use in that specific setting. These evaluations, however, rely on well-characterized reference sample that are often not available in resource-limited settings [46].

## Conclusions

Our study showed that both zoonotic diseases; Q fever and RVF are circulating among pastoralists and their livestock in Afar. Goats appeared to be the main species affected by Q fever and may be leading cause of transmission to humans. Cattle, on the other hand, appeared to be the main species affected by RVF with abortion events in this species possibly attributed to this viral infection.

To reduce the potential impact of large-scale outbreaks of both these pathogens in vulnerable pastoral regions, early detection and rapid response programs tailored to the mobility of pastoral communities are needed. This requires strengthening of integrated animal-human surveillance systems, including establishing suitable diagnostics, and improved communication between the human and animal health sectors.

## Acknowledgments

We thank AHRI for the logistical support and especially the AHRI laboratory team for their support of this project. We would also like to thank Silvia Cicconi from the Clinical Statistics and Data Management group and Jan Hattendorf from the One Health group, Swiss TPH for all their advice and support for the statistical analysis. A great thank you also goes to all participating pastoralists.

## Author Contributions

**Conceptualization:** Regina Bina Oakley, Rea Tschopp.

**Data curation:** Regina Bina Oakley, Ashenafi Gebregiorgis.

**Formal analysis:** Regina Bina Oakley, Rea Tschopp.

**Funding acquisition:** Daniel Henry Paris, Rea Tschopp.

**Investigation:** Regina Bina Oakley, Gizachew Gemechu, Ayinalem Alemu.

**Methodology:** Regina Bina Oakley.

**Project administration:** Regina Bina Oakley, Gizachew Gemechu, Rea Tschopp.

**Supervision:** Jakob Zinsstag, Daniel Henry Paris, Rea Tschopp.

**Validation:** Regina Bina Oakley.

**Visualization:** Regina Bina Oakley.

**Writing – original draft:** Regina Bina Oakley.

**Writing – review & editing:** Gizachew Gemechu, Ashenafi Gebregiorgis, Ayinalem Alemu, Jakob Zinsstag, Daniel Henry Paris, Rea Tschopp.

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
