## [Decision Letter · Decision Letter 0]

5 Mar 2024

Dear Dr Oakley,

Thank you very much for submitting your manuscript "Seroprevalence and risk factors for Q Fever and Rift Valley Fever in pastoralists and their livestock in Afar, Ethiopia: A One Health approach" for consideration at PLOS Neglected Tropical Diseases. As with all papers reviewed by the journal, your manuscript was reviewed by members of the editorial board and by several independent reviewers. In light of the reviews (below this email), we would like to invite the resubmission of a significantly-revised version that takes into account the reviewers' comments. 

We cannot make any decision about publication until we have seen the revised manuscript and your response to the reviewers' comments. Your revised manuscript is also likely to be sent to reviewers for further evaluation.

Sincerely,

Simon Rayner

Academic Editor

Justin Remais

Section Editor

Reviewer's Responses to Questions

**Key Review Criteria Required for Acceptance?**

**Methods**

-Are the objectives of the study clearly articulated with a clear testable hypothesis stated?

-Is the study design appropriate to address the stated objectives?

-Is the population clearly described and appropriate for the hypothesis being tested?

-Is the sample size sufficient to ensure adequate power to address the hypothesis being tested?

-Were correct statistical analysis used to support conclusions?

-Are there concerns about ethical or regulatory requirements being met?

Reviewer #1: (No Response)

Reviewer #2: (No Response)

Reviewer #3: Methods

-Are the objectives of the study clearly articulated with a clear testable hypothesis stated?

-> objectives yes, hypothesis no.

-Is the study design appropriate to address the stated objectives?

-> no. Study design, in pariclar sampling design could use some improvement. 

-Is the population clearly described and appropriate for the hypothesis being tested?

-> What is not comptelety clear is why a different number of household from each study site (the 5 sites) were sampled. Further comments within the point by point revisions document (attached word file). 

-Is the sample size sufficient to ensure adequate power to address the hypothesis being tested?

-> sample size was calculated suing appripriate tools, however clustering of the data (region-study site-hoseholds- humans/livestock) should have been considered. (More on this in the attached point by point revision.)

-Were correct statistical analysis used to support conclusions?

-> Yes

-Are there concerns about ethical or regulatory requirements being met?

-> Yes

**Results**

-Does the analysis presented match the analysis plan?

-Are the results clearly and completely presented?

-Are the figures (Tables, Images) of sufficient quality for clarity?

Reviewer #1: (No Response)

Reviewer #2: (No Response)

Reviewer #3: Results

-Does the analysis presented match the analysis plan?

-> in the analysis plan more detailed info on the so called risk factors should have been provided. 

-Are the results clearly and completely presented? (More on this in the attached point by point revision.)

-Are the figures (Tables, Images) of sufficient quality for clarity?

-> yes

**Conclusions**

-Are the conclusions supported by the data presented?

-Are the limitations of analysis clearly described?

-Do the authors discuss how these data can be helpful to advance our understanding of the topic under study?

-Is public health relevance addressed?

Reviewer #1: (No Response)

Reviewer #2: (No Response)

Reviewer #3: Conclusions

-Are the conclusions supported by the data presented?

-> mostly yes. There are conclusion on the stronger collaboration of human and animal sectors and while these are generell recommendations, the study it self and the data do not suggest anything on One Health governence level. 

-Are the limitations of analysis clearly described?

-> No, this is missing and should be added. 

-Do the authors discuss how these data can be helpful to advance our understanding of the topic under study?

-> Yes. 

-Is public health relevance addressed?

-> Yes.

**Editorial and Data Presentation Modifications?**

Reviewer #1: (No Response)

Reviewer #2: (No Response)

Reviewer #3: (No Response)

**Summary and General Comments**

Reviewer #1: Well written and timely manuscript.

Good use of One Health approach.

Good evidence and argument for tailoring epidemiological services to nomadic people.

Fills in some gaps in understudied RVFV-endemic region.

Some minor corrections suggested here:

Line 69 & throughout - use serial commas (i.e., should be “… humans, animals, and their environment …”)

Line 71 & throughout - Q fever (not Q fever)

Line 72 & throughout - Rift Valley fever (not Fever)

Line 73 - RVFV is also under-reported due to poor infrastructure, poor public education/engagement, and severe economic impact from OIE/WOAH trade bans.

Line 89 - clarify “animals”; redundant given rest of sentence?

Line 90 - through mosquitoes and other hematophagous arthropods

Line 91 - “RVF” should be “RVFV” — check throughout for appropriate use of RVF vs RVFV.

Line 93 - establish the use of RVFV throughout instead of “RVF virus”

Line 103 - should be “… migrating and searching for …”

Line 104 & throughout - suggest use either mobile or nomadic, not both; or clarify the terms

Line 101 & throughout - no need to capitalize compass directions

Line 107-111 — very good; should link anticipated results from statement in line 112-113 with suggested action to accomplish statement in line 107-111.

Line 122 - were should be was

Line 123 - should be “… representing five of the seven woredas, I.e., Amibara, Awash, …”

Line 127 - remove comma

Line 161 - should be “manufacturer’s”

Line 163 - should be “reported”

Line 182 - “was” should be “were”

Line 303 - separated by … what?

Line 324 - should be “… aged ≥49 years with C. burnetii seropositivity.”

Line 355 - Aedes should be italicized

Line 366 & throughout - be consistent with capitalization of Woredas

Line 377 - change “is” to “could be”

Line 392 - change “lower” to “shorter”

Line 399 - no comma after “drought”

Line 553 - missing authors on Ref #32

Reviewer #2: (No Response)

Reviewer #3: Please refere to the line by line revision document (word file) sugegstion improvement of the manuscript.

PLOS authors have the option to publish the peer review history of their article (what does this mean?). If published, this will include your full peer review and any attached files.

Reviewer #1: No

Reviewer #2: No

Reviewer #3: No
---

## [Decision Letter · Decision Letter 1]

24 Jun 2024

Dear Dr Oakley,

Thank you very much for submitting your manuscript "Seroprevalence and risk factors for Q fever and Rift Valley fever in pastoralists and their livestock in Afar, Ethiopia: A One Health approach" for consideration at PLOS Neglected Tropical Diseases. As with all papers reviewed by the journal, your manuscript was reviewed by members of the editorial board and by several independent reviewers. The reviewers appreciated the attention to an important topic. Based on the reviews, we are likely to accept this manuscript for publication, providing that you modify the manuscript according to the review recommendations.

Sincerely,

Simon Rayner

Academic Editor

Justin Remais

Section Editor

Reviewer's Responses to Questions

**Key Review Criteria Required for Acceptance?**

**Methods**

-Are the objectives of the study clearly articulated with a clear testable hypothesis stated?

-Is the study design appropriate to address the stated objectives?

-Is the population clearly described and appropriate for the hypothesis being tested?

-Is the sample size sufficient to ensure adequate power to address the hypothesis being tested?

-Were correct statistical analysis used to support conclusions?

-Are there concerns about ethical or regulatory requirements being met?

Reviewer #2: (No Response)

Reviewer #3: Thank you to the authors for their accomplished efforts in revising their manuscript accoding to reviewer guidance givven. I blieve the mansucript has improved technically, scientifically and in readability. Thank you also fo providing detailed explanation on where amanemends weren´t possible authors hae agrumented otherwise. This helped to understand the justufucation for sthe study conducted.

**Results**

-Does the analysis presented match the analysis plan?

-Are the results clearly and completely presented?

-Are the figures (Tables, Images) of sufficient quality for clarity?

Reviewer #2: (No Response)

Reviewer #3: (No Response)

**Conclusions**

-Are the conclusions supported by the data presented?

-Are the limitations of analysis clearly described?

-Do the authors discuss how these data can be helpful to advance our understanding of the topic under study?

-Is public health relevance addressed?

Reviewer #2: (No Response)

Reviewer #3: (No Response)

**Editorial and Data Presentation Modifications?**

Reviewer #2: (No Response)

Reviewer #3: (No Response)

**Summary and General Comments**

Reviewer #2: (No Response)

Reviewer #3: (No Response)

PLOS authors have the option to publish the peer review history of their article (what does this mean?). If published, this will include your full peer review and any attached files.

Reviewer #2: No

Reviewer #3: No

Figure Files:

Data Requirements:

Reproducibility:

References

---

## [Editor Report · Decision Letter 2]

22 Jul 2024

Dear Dr Oakley,

We are pleased to inform you that your manuscript 'Seroprevalence and risk factors for Q fever and Rift Valley fever in pastoralists and their livestock in Afar, Ethiopia: A One Health approach' has been provisionally accepted for publication in PLOS Neglected Tropical Diseases.

Best regards,

Simon Rayner

Academic Editor

Justin Remais

Section Editor

---

## [Editor Report · Acceptance letter]

2 Aug 2024

Dear Dr Oakley,

We are delighted to inform you that your manuscript, "Seroprevalence and risk factors for Q fever and Rift Valley fever in pastoralists and their livestock in Afar, Ethiopia: A One Health approach," has been formally accepted for publication in PLOS Neglected Tropical Diseases.

Best regards,

Shaden Kamhawi

co-Editor-in-Chief

Paul Brindley

co-Editor-in-Chief
